# CT-GAT: Cross-Task Generative Adversarial Attack based on Transferability

**Minxuan Lv**[1,2], **Chengwei Dai**[1,2], **Kun Li**[1*], **Wei Zhou**[1], **Songlin Hu**[1]

[1]Institute of Information Engineering, Chinese Academy of Sciences
[2]School of Cyber Security, University of Chinese Academy of Sciences
{lvminxuan, daichengwei, likun2, zhouwei, husonglin}@iie.ac.cn

## Abstract

Neural network models are vulnerable to adversarial examples, and adversarial transferability further increases the risk of adversarial attacks. Current methods based on transferability often rely on substitute models, which can be impractical and costly in real-world scenarios due to the unavailability of training data and the victim model's structural details. In this paper, we propose a novel approach that directly constructs adversarial examples by extracting transferable features across various tasks. Our key insight is that adversarial transferability can extend across different tasks. Specifically, we train a sequence-to-sequence generative model named **CT-GAT** (**C**ross-**T**ask **G**enerative **A**dversarial **AT**tack) using adversarial sample data collected from multiple tasks to acquire universal adversarial features and generate adversarial examples for different tasks. We conduct experiments on ten distinct datasets, and the results demonstrate that our method achieves superior attack performance with small cost. You can get our code and data at: https://github.com/xiaoxuanNLP/CT-GAT

## 1 Introduction

Neural network-based natural language processing (NLP) is increasingly being applied in real-world tasks(Oshikawa et al., 2018; Xie et al., 2022; OpenAI, 2023). However, neural network models are vulnerable to adversarial examples(Papernot et al., 2016; Samanta and Mehta, 2017; Liu et al., 2022). Attackers can bypass model-based system monitoring by intentionally constructing adversarial samples to fulfill their malicious objectives, such as propagating rumors or hate speech. Even worse, researchers have discovered the phenomenon of adversarial transferability, where adversarial examples can propagate across models trained on the same or similar tasks(Papernot et al., 2016; Jin et al., 2019; Yuan et al., 2020; Datta, 2022; Yuan

---

*Kun Li is the corresponding author.

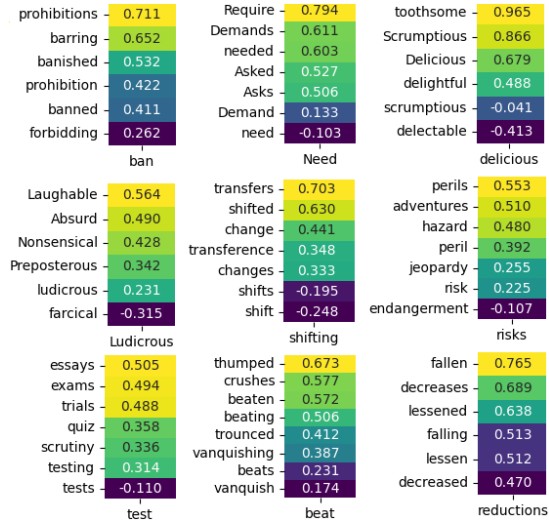

Figure 1: Heatmap of the Highly-transferable Adversarial Word replacement Rules (HAWR) on all datasets in TCAB. The bottom and left words of the heatmap represent the replaced words and synonyms respectively. The values in the figure are derived using this rule, with higher values indicating greater transferability. The heatmap in the figure provides an intuitive observation that a replaced word has multiple replacements with high transferability.

et al., 2021). This transferability enables attackers to target victim models using adversarial examples crafted by substitute models.

The majority of existing research on adversarial transferability involves training one or more substitute models that perform identical or similar tasks to the victim model (Datta, 2022; Yuan et al., 2021). However, due to the constraints of the black box scenario, the attacker lacks access to the training data and structural details of the victim model, making it exceedingly challenging to train comparable substitute models. Consequently, it becomes arduous to achieve a satisfactory success rate in adversarial attacks solely through transferability between models based on the same task.

To mitigate the above challenges, we introduce

a method that directly constructs adversarial examples by extracting transferable features across various tasks, without the need for constructing task-specific substitute models. Our key insight is that adversarial transferability is not limited to models trained on the same or similar tasks, but rather extends to models across different tasks. There are some observations that support our idea. For instance, we discovered that adversarial word substitution rules offer a variety of highly transferable candidate replacement words. As illustrated in Figure 1, a larger pool of candidate words can be utilized to generate a diverse set of highly transferable adversarial samples, thereby compensating for the shortcomings of previous methods that relied on greedy search. In particular, we train a sequence-to-sequence generative model **CT-GAT** (**C**ross-**T**ask **G**erative Adversarial **AT**tack) using adversarial sample data obtained from multiple tasks. Remarkably, we find that the generated adversarial sample can effectively attack test tasks, even in the absence of specific victim model information.

We conduct attack experiments on five security-related NLP tasks across ten datasets, adhering to the Advbench(Chen et al., 2022) paradigm of Security-oriented Natural Language Processing (SoadNLP), a framework that mirrors real-world application scenarios more closely. And the experiments demonstrate that our method achieves the best attack performance while using a relatively small number of queries.

To summarize our contributions to this paper are as follows:

- We introduce an approach that leverages texts to learn an effective, transferable adversarial text generator.

- Our method can combine adversarial features from different tasks to enhance the adversarial effect.

- We test the effectiveness of our method in the decision-based black-box scenario. The experiments demonstrate that our method achieves the best attack performance while using a relatively small number of queries.

## 2 Related Work

In this section, we review the recent works on adversarial attacks, with a focus on transferability-based approaches.

White-box attacks are prominent methods in attack methods, which can obtain high attack success rate and low distortion adversarial data with few access times. Typical white-box attack methods included CWBA(Liu et al., 2022), GBDA(Guo et al., 2021). The effectiveness of obtaining adversarial samples efficiently in these scenarios stemmed from the fact that attackers have comprehensive access to information, including the model's structure, parameters, gradients, and training data.

Black-box attacks assume that the attacker only has knowledge of the confidence or decision of the target's output, including query-based attacks and transfer-based attacks. Query-based attack methods include PWWS(Ren et al., 2019), TextBugger(Li et al., 2018), Hotflip(Ebrahimi et al., 2018), etc. These methods often required hundreds of queries to successfully obtain low-distortion adversarial samples, and the attack effect may have been worse in the decision-based scenario.

Adversarial transferability has been observed across different models(Szegedy et al., 2014). This poses a threat to the security of decision-based black-box models. Attackers employ white-box attacks to obtain adversarial samples on substitute models. This method is prone to overfitting, resulting in limited transferability. To address this issue, some methods proposed using mid-layer features instead of the entire model to obtain more generalized adversarial samples(Wang et al., 2023). However, this approach has not been widely applied in the text adversarial domain. Model aggregation methods were primarily used in the text domain, although the overall research in this area was not extensive. Additionally, there were approaches that extracted adversarial perturbation rules as a basis for adversarial features(Yuan et al., 2021). This rule-based approach provided insights into leveraging the distinctive characteristics of different adversarial samples.

## 3 Methodology

### 3.1 Textual Adversarial Attack

For a natural language classifier $f$, it can accurately classify the original input text into the label $y_{true}$ based on the maximum a priori probability:

$$arg \max_{y_i \in Y} f(y_i|x) = y_{true} \qquad (1)$$

Where $y_{true}$ represents the ground truth label of input $x$. To fool the classifier, the attacker in-

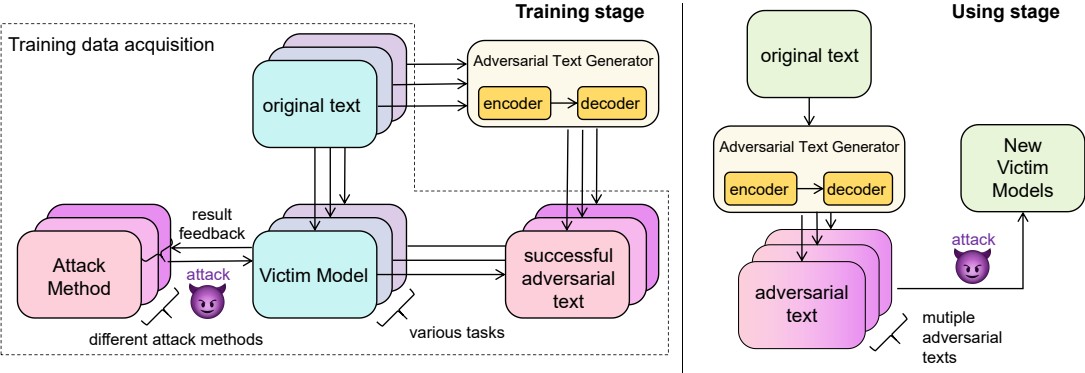

Figure 2: Overview of the training process and application of our approach. (Training stage) We initially launch attacks tailored to different tasks, employing one or more attack strategies until successful adversarial examples are obtained. We then use the original samples and the successful adversarial examples as training data for the adversarial example generator, learning potential common patterns within these training data. (Using stage) The trained generator can produce a large number of adversarial examples without the need for feedback from the victim model, integrating various attack methods. The high transferability learned enables it to successfully attack victim models that were not encountered during the training phase.

troduces a perturbation $\delta$ to the original input $x$, thereby creating an adversarial sample $x_{adv}$.

$$arg \max_{y_i \in Y} f(y_i|x_{adv}) \neq y_{true}, x_{adv} = x + \delta \quad (2)$$

Typically, the perturbation $\delta$ is explicitly constrained, such as by setting a limit on the number of modifications(Li et al., 2018) or by imposing semantic constraints(Gan and Ng, 2019). In other words, the adversarial sample $x_{adv}$ is intended to remain similar to the original sample $x$. In our approach, this constraint is not explicitly imposed but is implicitly reflected in the training data. The adversarial examples in training data are obtained under explicit constraint conditions.

### 3.2 Transferability-Based Generator for Perturbations

To effectively learn the distributional characteristics from original sentences to adversarial sentences and generate diverse adversarial samples, we choose to utilize the encoder-decoder architecture of Transformer(Vaswani et al., 2017). The encoder uses self-attention mechanism, which can comprehensively extract the features and context information of the original sentence. The decoder can handle variable-length problems, enabling the learning of complex attacks such as character-level attacks and phrase-level attacks. The use of the encoder-decoder architecture allows us to combine the strengths of both components. The learning objective of this task can be formalized as maximizing

the following likelihood:

$$L(U, \hat{U}) = \sum_i log P(u_i|u_{<i}, X, \Theta) \quad (3)$$

We first train victim models on multiple datasets across different tasks, each using a distinct model structure. Successful adversarial samples are obtained from various attack methods on each victim model, and these samples are then mixed to serve as training data for the generator. We aim to learn features with high transferability across tasks and models through this data, and to integrate various attack methods to generate a more diverse set of adversarial samples. Due to the efficiency issues with traditional attack methods, where generating a large number of adversarial samples is time-consuming, we directly utilize the TCAB(Asthana et al., 2022) dataset as training data.

The TCAB dataset consists of two tasks with a total of six datasets: hate speech detection (**Hatebase**(Davidson et al., 2017), **Civil Comments**[1], and **Wikipedia**(Wulczyn et al., 2016; Dixon et al., 2018)) and sentiment analysis (**SST-2**(Socher et al., 2013), **Climate Change**[2], and **IMDB**(Maas et al., 2011)).

The TCAB dataset consists of adversarial samples at different granularities, including word-level adversarial samples (Genetic(Alzantot et al., 2018), TextFooler etc.), character-level adversarial samples (DeepWordBug(Gao et al., 2018), HotFlip

---

[1] https://www.kaggle.com/c/jigsaw-unintended-bias-in-toxicity-classification
[2] https://www.kaggle.com/edqian/twitter-climate-change-sentiment-dataset

etc.), and mixed-granularity adversarial samples (TextBugger, Viper(Eger et al., 2019), etc.). These methods include both white-box attacks and black-box attacks. The adversarial samples of each task are obtained by attacking BERT, RoBERTa, and XLNet respectively. Our generation network is trained across diverse datasets, models, and attack methods to learn universal adversarial features.

We don't strictly define the bounds of perturbations as previous methods did because it is implicitly reflected in the training data. In the adversarial sample generation stage, we can control the perturbation degree of the adversarial samples by adjusting the temperature parameter in the model logits.

## 4 Experiments and Results

In this chapter, we evaluate the performance of CT-GAT in the decision-based black-box scenario, using the security-oriented NLP benchmark Advbench. To demonstrate the high interpretability of CT-GAT in practical settings, we also conducted a manual evaluation.

### 4.1 Implementation

**Victim Model** BERT is a widely recognized pre-trained transformer that is highly representative in natural language understanding (NLU) tasks. We chose to fine-tune the BERT model as the victim model for all tasks.

**Generation Model** We choose BART (Lewis et al., 2019) as our generation model. BART learns to generate text that is semantically similar to the original data by trying to recover it from its perturbed version. This enables BART to fully learn the similarities and differences between different word meanings and to have the ability to preserve semantic invariance. Moreover, BART is trained on a large amount of text and possesses the ability to analyze parts of speech, which allows it to further learn complex patterns of replacement during fine-tuning.

### 4.2 Baseline

Our baseline method follows the experimental setup of Advbench. We use the NLP attack package OpenAttack [3] (Zeng et al., 2020) to implement some common attacks. We implement the rocket attack using the source code provided by

---

[3] https://github.com/thunlp/OpenAttack

Advbench. Specifically, the attack methods we employ include (1)TextFooler, (2)PWWS, (3)BERT-Attack(Li et al., 2020), (4)SememePSO(Zang et al., 2020), (5)DeepWordBug(Gao et al., 2018), (6)ROCKET(Chen et al., 2022). Furthermore, we train CT-GAT$_{word}$ using word-level adversarial samples from the TCAB to examine if it learns imperceptible perturbation features, as character-level attacks often introduce grammar errors and increase perplexity.

### 4.3 Datasets

We evaluate CT-GAT on nine datasets from Advbench, which include tasks such as misinformation, disinformation, toxic content, spam, and sensitive information detection (excluding the HSOL dataset(Davidson et al., 2017)). Since the HSOL dataset and the dataset used to train our adversarial sample generator are the same, we substitute it with the Founta dataset (Founta et al., 2018) in this study.

### 4.4 Evaluation metrics

To evaluate the effectiveness of CT-GAT, we need to measure the attack success rate, query count, perturbation degree, and text quality. (1) ASR is defined as the percentage of successful adversarial samples. (2)The query count is defined as the average number of queries required to make adversarial samples. (3)The perturbation degree is measured by the Levenshtein distance. (4)Text quality is measured by the relative increase in perplexity (% PPL) and the absolute increase in the number of grammer errors ($\Delta I$). (5)Semantic similarity is an evaluative metric quantifying the extent to which the meaning of the original sentence is preserved post-perturbation. We represent this using the cosine similarity of Universal Sentence Encoder (USE) vectors.

### 4.5 Experimental Results

**ASR** and **Query Count** The ASR and query count results are shown in Table 1. Our method CT-GAT achieves excellent results on all tasks and datasets. CT-GAT achieves the highest ASR and the least number of queries on majority of datasets. On a few datasets, even though it do not achieve the highest ASR, it achieved the second highest ASR. With only 2.3%-5.6% lower than the highest ASR method, the number of queries was reduced by 31.9%-91.2%. The ASR of CT-GAT did not achieve optimal results on the EDENCE, FAS, and

| Task | Misinformation | | Disinformation | | Toxic | | Spam | | Sensitive Information | |
|---|---|---|---|---|---|---|---|---|---|---|
| Method \| Dataset | LUN | | Amazon-LB | | Founta | | SpamAssassin | | EDENCE | |
| | ASR(%)↑ | Query↓ | ASR(%)↑ | Query↓ | ASR(%)↑ | Query↓ | ASR(%)↑ | Query↓ | ASR(%)↑ | Query↓ |
| TextFooler | 0.4 | 1294.38 | 9.0 | 740.42 | 52.7 | 67.46 | 0.2 | 961.88 | 23.9 | 94.67 |
| PWWS | 1.3 | 1707.19 | 18.8 | 1019.91 | 61.0 | 113.89 | 0.3 | 1308.50 | 46.0 | 129.68 |
| BERT-Attack | 7.0 | 3966.60 | 43.0 | 1625.37 | 77.0 | 109.52 | 2.2 | 4336.18 | **90.3** | 140.98 |
| SememePSO(maxiter=100) | 0.9 | 2020.85 | 23.8 | 1627.97 | 79.5 | 209.59 | 0.9 | 1945.74 | 79.6 | 231.17 |
| DeepWordBug(power=5) | 0.1 | 287.04 | 9.3 | 162.37 | 40.7 | 21.78 | 0.1 | 263.84 | 22.9 | 26.06 |
| DeepWordBug(power=25) | 0.2 | 287.04 | 12.4 | 162.41 | 65.5 | 22.03 | 0.0 | 263.84 | 79.9 | 26.63 |
| ROCKET | 7.2 | 300.38 | 38.7 | 218.69 | 97.0 | **5.03** | 1.1 | 60.09 | 84.5 | 20.93 |
| CT-GAT$_{word}$(our) | 1.9 | 98.58 | 19.5 | 83.44 | 90.3 | 14.28 | 3.1 | 59.45 | 53.0 | 58.91 |
| CT-GAT(our) | **10.0** | **92.69** | **56.2** | **54.51** | **99.3** | 39.02 | **14.8** | **59.36** | 88.0 | **19.08** |
| Acc.(%) | 99.2 | | 92.1 | | 93.1 | | 99.7 | | 97.8 | |

| | SATNews | | CGFake | | Jigsaw2018 | | Enron | | FAS | |
|---|---|---|---|---|---|---|---|---|---|---|
| | ASR(%)↑ | Query↓ | ASR(%)↑ | Query↓ | ASR(%)↑ | Query↓ | ASR(%)↑ | Query↓ | ASR(%)↑ | Query↓ |
| TextFooler | 2.9 | 1889.32 | 18.2 | 360.13 | 12.5 | 201.72 | 0.1 | 682.40 | 17.4 | 130.73 |
| PWWS | 1.2 | 2565.37 | 69.0 | 489.78 | 20.2 | 268.87 | 0.0 | 928.58 | 36.5 | 177.18 |
| BERT-Attack | 30.6 | 5102.34 | 94.6 | 400.61 | 40.4 | 450.67 | 1.4 | 2954.86 | **92.4** | 305.59 |
| SememePSO(maxiter=100) | 4.2 | 2217.34 | 67.2 | 689.46 | 51.9 | 539.25 | 1.0 | 1724.70 | 61.4 | 506.82 |
| DeepWordBug(power=5) | 2.5 | 430.59 | 41.7 | 75.00 | 35.9 | 45.71 | 0.0 | 182.27 | 40.8 | 39.91 |
| DeepWordBug(power=25) | 1.9 | 430.59 | 68.8 | 75.28 | 57.6 | 45.92 | 0.0 | 182.27 | 77.6 | 40.27 |
| ROCKET | 4.4 | 324.30 | 97.2 | 37.11 | 64.2 | 78.85 | **6.5** | 56.92 | 82.0 | 52.77 |
| CT-GAT$_{word}$(our) | 1.5 | 97.25 | 69.25 | 43.09 | 67.0 | 40.75 | 4.0 | 38.97 | 42.0 | 67.60 |
| CT-GAT(our) | **39.0** | **69.59** | **99.2** | **6.38** | **75.2** | **34.93** | 4.2 | **38.79** | 86.8 | **26.92** |
| Acc.(%) | 96.6 | | 99.1 | | 95.7 | | 99.3 | | 96.3 | |

Table 1: The results of attack performance for various attack methods in decision-based black-box scenarios. ASR stands for the attack success rate. Query signifies the average number of launching a successful adversarial attack. Acc represents the accuracy of the victim model on the test set. A higher ASR and a lower Query indicate a more effective attack. The bolded part indicates the best result, and the underlined part indicates the second best result.

Enron datasets, which is related to the degree of perturbation. This can be seen more clearly in Table 3, where our method has half the perturbation rate of the method with the highest ASR on these three datasets, which indicates that the imperceptibility learned from the data is constraining our perturbation method. Although we can increase the ASR by increasing the sampling temperature, this may greatly affect the readability and change the original sentence meaning.

To further evaluate the effectiveness of CT-GAT, we conduct additional experiments on two datasets, Jigsaw2018[4] and EDENCE, which have similar average query counts to the baseline. We conduct attacks on the model under strict constraints on the number of access queries, as shown in Figure 3. The differences in our replication of the ROCKET method are large, therefore they are not included in the graph for comparison. We observe that CT-GAT consistently outperforms other methods and achieves higher attack success rates even at extremely low query counts.

**Perturbation Degree**, **Text Quality** and **Consistency** From Table 3, we can observe the level of perturbation and text quality of CT-GAT, and the baseline. Considering the previous attack results, the CT-GAT$_{word}$ method demonstrates good performance in terms of perturbation rate and the number of introduced grammar errors while maintaining satisfactory attack effectiveness. This may be attributed to the model's ability to discern parts of speech and assess the importance of substitute words. To some extent, this demonstrates the capability of CT-GAT to learn the distribution of adversarial examples and the implicit constraint of imperceptibility.

However, perplexity does not necessarily reflect the real reading experience. For example, the ROCKET method uses a large number of repeated words or letters inserted into the original sentence, which makes the model easy to predict the words and reduces the perplexity. CT-GAT, due to the introduction of character-level attacks, does not seem outstanding in terms of grammatical errors and fluency. But in fact, we may not encounter too many obstacles when reading in practice. We follow the description of the Advbench, which emphasizes that the purpose of the attack is to propagate harmful information. We design human evaluations to measure the effectiveness of the method.

In terms of consistency, the cosine similarity of our method is at a relatively high level, indicating

[4]https://www.kaggle.com/c/jigsaw-toxic-comment-classification-challenge

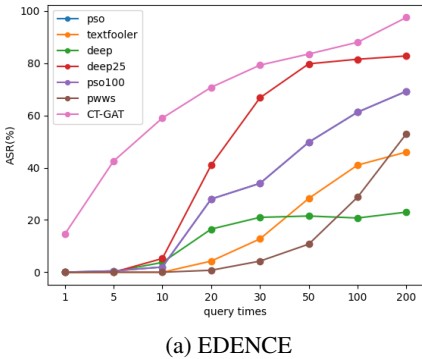

(a) EDENCE

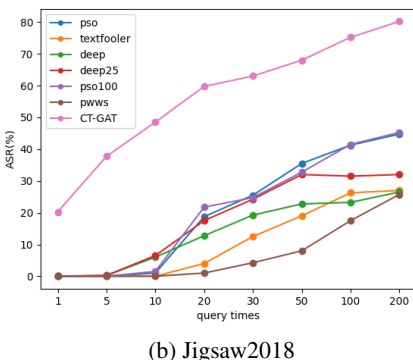

(b) Jigsaw2018

Figure 3: Attack success rate under restriction of maximum query number in (a)EDENCE, (b)Jigsaw2018

that the sentences before and after the perturbation by our method maintain good semantic consistency. To further analyze semantic consistency, we adopt a human evaluation approach.

| Model | 2 | 1 | 0 | 2+1 | succ |
|---|---|---|---|---|---|
| Founta | 64.5 | 23.5 | 12.0 | 88.0 | 83.5 |
| FAS | 56.0 | 39.0 | 5.0 | 95.0 | 96.5 |

Table 2: Human evaluation is conducted on two tasks: hate speech detection and sensitive information detection. The evaluation uses the following scale: **0**=meaning changed, **1**=partially preserved, **2**=essentially the same. **Succ** indicates whether the adversarial samples retain their original labels according to the human evaluation.

**Human Evolution** Automatic metrics provide insights into the quality of the text. To verify whether adversarial examples are truly effective in real-world applications, we adopt a manual evaluation approach.

We choose two tasks, hate speech detection and sensitive information detection, because these tasks have clear definitions and are easy to evaluate manually. We select the Founta dataset and the FAS

dataset, each consisting of 40 pairs of original texts and adversarial texts. For each pair of samples, we ask 5 evaluators to assess the following aspects sequentially: (1) whether the adversarial sample conveys malicious intent (or contains sensitive information), and (2) the degree to which the intended meaning of the original sample and the adversarial sample aligns, measured on a scale of 0-2. The label of 0 indicates no relevance at all, 1 indicates partial conveyance of the original sentence's meaning, and 2 indicates substantial conveyance of the original sentence's meaning.

The results of the human evaluation are presented in Table 2. Labels of 1 or 2 are considered meaningful. It can be observed that CT-GAT can largely preserve the original meaning and effectively convey malicious intent.

### 4.6 Case Study

We select three successful adversarial examples from EDENCE and Amazon-LB. It can be observed from Table 4. The perturbation patterns of these adversarial samples originate from previous attack methodologies, effectively learning and integrating prior adversarial features at both the character and word levels. This includes tactics such as word segmentation, similar character substitution, and synonym replacement.

## 5 Further Analysis

In this chapter, our objective is to thoughtfully examine the areas where prior research could possibly be further enriched, particularly in terms of transferability. Simultaneously, we will elucidate the factors that enable our approach to effectively harness certain transferable attributes.

### 5.1 Cross-task Adversarial Features

In this section, we aim to validate our hypothesis that leveraging adversarial features from different tasks can enhance the attack success rate for the current task. Based on previous research(Yuan et al., 2021), we can formulate the extraction of adversarial features as a perturbation substitution rule. We propose **C**ross-task **A**dversarial **W**ord **R**eplacement rules (CAWR) to expand the rules in order to extract adversarial features across different tasks. The specific algorithm details can be found in Appendix A.

We partition the TCAB(Asthana et al., 2022) dataset into three subsets for our study: the **Hate-**

| Task | Misinformation | | | | Toxic | | | | Sensitive Information | | | |
|---|---|---|---|---|---|---|---|---|---|---|---|---|
| Method \| Dataset | Amazon-LB | | | | Founta | | | | EDENCE | | | |
| | Levenstein↓ | ΔI↓ | %PPL↓ | USE↑ | Levenstein↓ | ΔI↓ | %PPL↓ | USE↑ | Levenstein↓ | ΔI↓ | %PPL↓ | USE↑ |
| TextFooler | 25.68 | 0.57 | 1.79 | **0.95** | 18.29 | 3.15 | 1.21 | **0.87** | 11.86 | 0.10 | 2.51 | **0.87** |
| PWWS | 95.89 | 1.27 | 3.69 | 0.84 | 20.38 | 2.93 | 1.65 | 0.81 | 30.06 | 0.21 | 5.84 | 0.87 |
| BERT-Attack | 117.31 | 0.41 | 6.60 | 0.81 | 23.41 | 2.56 | 1.69 | 0.76 | 24.78 | 0.07 | 3.69 | 0.74 |
| SememePSO(maxiter=100) | 28.58 | 0.64 | 1.93 | 0.92 | 21.37 | 3.08 | 1.62 | 0.80 | 14.71 | 0.09 | 3.35 | 0.86 |
| DeepWordBug(power=5) | **18.86** | 0.33 | 4.19 | 0.82 | **8.05** | 4.23 | 1.13 | 0.73 | **6.66** | -0.02 | 5.13 | 0.57 |
| DeepWordBug(power=25) | 34.43 | 0.23 | 4.27 | 0.32 | 17.96 | 11.43 | 0.19 | 0.24 | 19.57 | **-0.04** | 2.41 | 0.08 |
| ROCKET | 82.93 | 1.01 | 1.81 | 0.71 | 367.67 | 15.48 | **-0.98** | 0.05 | 97.95 | 5.01 | **0.34** | 0.32 |
| CT-GAT$_{word}$(our) | 50.51 | **-1.00** | 0.12 | 0.90 | 18.02 | 0.29 | 0.61 | 0.81 | 13.27 | 0.10 | 0.63 | 0.87 |
| CT-GAT(our) | 82.96 | 11.67 | **-0.15** | 0.73 | 18.49 | **0.27** | 0.61 | 0.80 | 12.90 | 2.93 | 1.69 | 0.76 |
| Method \| Dataset | Amazon-LB | | | | Founta | | | | EDENCE | | | |
| | Levenstein↓ | ΔI↓ | %PPL↓ | USE↑ | Levenstein↓ | ΔI↓ | %PPL↓ | USE↑ | Levenstein↓ | ΔI↓ | %PPL↓ | USE↑ |
| TextFooler | 16.62 | 0.23 | 2.12 | **0.93** | 14.64 | 0.09 | 1.21 | **0.84** | 13.12 | 0.05 | 2.45 | **0.90** |
| PWWS | 101.47 | 1.11 | 4.69 | 0.80 | 52.76 | 0.38 | 4.19 | 0.79 | 48.93 | 0.23 | 5.04 | 0.84 |
| BERT-Attack | 82.85 | 0.48 | 11.41 | 0.79 | 30.65 | 0.05 | 3.33 | 0.77 | 53.20 | 0.10 | 6.12 | 0.75 |
| SememePSO(maxiter=100) | 23.77 | 0.31 | 3.15 | 0.86 | 18.23 | 0.11 | 3.42 | 0.76 | 15.55 | 0.03 | 2.37 | 0.82 |
| DeepWordBug(power=5) | **10.01** | 0.05 | 5.16 | 0.83 | **10.79** | 0.03 | 3.45 | 0.60 | **6.65** | -0.02 | 3.64 | 0.58 |
| DeepWordBug(power=25) | 29.49 | 0.00 | 4.13 | 0.26 | 20.49 | -0.05 | 2.27 | 0.25 | 18.33 | **-0.03** | 2.44 | 0.15 |
| ROCKET | 38.27 | 1.03 | **1.66** | 0.60 | 1084.44 | 44.99 | **-0.96** | 0.05 | 97.95 | 5.01 | 0.34 | 0.41 |
| CT-GAT$_{word}$(our) | 64.08 | **-0.85** | 3.01 | 0.86 | 20.47 | 0.05 | 0.11 | 0.76 | 23.01 | 0.40 | **0.25** | 0.84 |
| CT-GAT(our) | 42.46 | 6.43 | 9.40 | 0.79 | 16.28 | 2.54 | 1.03 | 0.70 | 25.70 | 6.84 | 1.16 | 0.69 |

Table 3: The results of attacking performance and adversarial samples' quality. $\Delta I$ represents the relative increase in the number of grammatical errors, %PPL represents the relative increase rate of perplexity.

| | |
|---|---|
| Original Sentence | it is true that the bush administration refused to help enron stave off bankruptcy. |
| Adversarial Sentence | it is tru e that the bush administration refused to assistance en ron stave off bankruptcy. |
| Original Sentence | was able to withstand the pressure. |
| Adversarial Sentence | was aƀle to withstɑnd the pressure. |
| Original Sentence | great consistency. i love essie. beautiful color too! |
| Adversarial Sentence | ǵřeat cônsîsteňcy. i love êssîe. beaǔtîfu, coloř too! |

Table 4: Adversarial samples crafted by CT-GAT. The red parts correspond to the modified words or characters (blue parts) in the original sentence.

**base** dataset, the hate speech detection datasets (comprising **Hatebase**, **Civil Comments**), and a mixed dataset that combines sentiment detection with the previous abuse detection datasets and additional datasets (**SST-2**, **Climate Change**, and **IMDB**). We extract rules from these subsets and attack models on Jigsaw2018, Founta(Founta et al., 2018), and tweet[5]. Table 5 shows the results. It can be inferred that, overall, regardless of whether the adversarial sample construction method is simple and rule-based or based on model generation, the amalgamation of different task methods tends to result in a higher rate of successful attacks, or in other terms, enhanced transferability.

## 5.2 Selection of Words

In this subsection, we discuss some evidence that CT-GAT can learn transferable adversarial features

[5]https://github.com/sharmaroshan/Twitter-Sentiment-Analysis

directly from the text. The process of generating adversarial samples can be considered a search problem(Zang et al., 2020), where we aim to find the vulnerable words and the optimal substitute words.

**Vulnerable Words** The selection of vulnerable words is the process of choosing words to be replaced. We aim to investigate whether different models exhibit similar patterns of vulnerable (or important) word distributions, which could be one of the manifestations of transferability across different models for the same task. We measure the vulnerability of words by sequentially masking them in a sentence and comparing the difference in confidence scores of the correct class before and after masking the input to the model.

We rank the vulnerability levels of the same sentence across different models and compare the Jaccard similarity of the selected top words to assess the consistency among different models. As shown in Table 6, We find some consistency both between

| ASR | Jigsaw | Founta | tweet |
|---|---|---|---|
| Hatebase | 0.116 | 0.195 | 0.449 |
| Toxic domain | 0.131 | 0.217 | 0.507 |
| All datasets | 0.132 | 0.221 | 0.552 |

(a) CAWR

| ASR | Jigsaw | Founta | tweet |
|---|---|---|---|
| Hatebase | 0.620 | 0.937 | 0.855 |
| Toxic domain | 0.705 | 0.948 | 0.468 |
| All datasets | 0.738 | 0.958 | 0.520 |

(b) CT-GAT$_{word}$

Table 5: The CAWR (a) and CT-GAT$_{word}$ (b) method demonstrates adversarial transferability using features from different domains. Specifically, it utilizes one dataset, three datasets, and six datasets successively from top to bottom.

| Model | Jaccard |
|---|---|
| tweets & Founta | 0.272 |
| tweets & hatebase | 0.519 |
| Founta & hatebase | 0.318 |

(a) Different training data

| Architecture | Jaccard |
|---|---|
| BERT & RoBERTa | 0.508 |
| BERT & XLNet | 0.508 |
| RoBERTa & XLNet | 1.000 |

(b) Different architectures

Table 6: Models using (a) training datasets and (b) different architectures are evaluated by calculating the Jaccard similarity for the top 20% percent vulnerable words. The Jaccard similarity calculated on a randomly sampled set of the top 20% words is 0.125

different model architectures and among training data from similar domains.

We extract the substitution rules from CT-GAT$_{word}$ for generating adversarial samples and find that the average perturbation rate is 16.6%. The proportion of the words replaced by CT-GAT$_{word}$ that belong to the top 20% vulnerable word set is 36%. These results indicate that it is possible to learn the vulnerable words of a sentence directly from the adversarial text.

**Substitution preference** Figure 1 presents the heatmap of transferability metrics for replacement rules, as derived through the HAWR algorithm by Yuan et al. (2021). Upon intuitive observation, it is apparent that certain replaced words possess multiple highly transferable replacements. Simultaneously, we compute the Gini inequality coefficient by tallying the number of word replacement rules in successful adversarial samples. Through statistical analysis of the replacement preferences

for each substituted word, we derived a Gini imbalance coefficient of 0.161. This suggests that the process of word replacement adheres to a discernible pattern, rather than manifesting as chaotic, and further demonstrates a certain level of diversity, which aligns with our observations from the heatmap. The prior application of greedy methods to select the most transferable replacement rules exhibits certain constraints, whereas the technique of employing a network for searching can effectively discern diverse patterns.

The previously mentioned transferable elements can all be represented in the statistical rules of the text. This offers the potential for employing neural network methodologies to learn transferable characteristics from adversarial samples.

### 5.3 Defense

Learning to map from adversarial samples back to the original sample distribution is relatively easier compared to the inverse process. To further validate the ability to learn adversarial sample distributions from text, we conduct defensive experiments using generative models. We use adversarial samples as input and the original samples as predictions. Generative models can serve as a data cleansing method by cleaning the input text before feeding it into the victim model. Appendix B presents the defensive experiments conducted on two victim models trained using CGFake and Hatebase datasets.

Overall, using this method can effectively reduce the success rate of attackers and increase the cost of queries. CT-GAT demonstrates excellent defense against character-level attack methods like DeepWordBug, indicating the presence of strong patterns in character-level attacks. However, the performance against word-level attacks is not particularly outstanding. This mainly occurs because the attack method gradually intensifies its attack magnitude after a failed attack. This surpasses the defense model's capability, which is trained on imperceptible samples.

### 6 Conclusion

In this article, we propose a cross-task generative adversarial attack (CT-GAT), which is based on our insight that adversarial transferability exists across different tasks. Our method solves the problem of needing to train a substitute model for the victim model. Experiment demonstrates that our method

not only significantly increases the attack success rate, but also drastically reduces the number of required queries, thus underscoring its practical utility in real-world applications.

## Limitations

Our method has the following limitations: (1) CT-GAT was trained on a limited dataset. While it has demonstrated effectiveness within the scope of our experiments, the ability to generalize to broader, real-world scenarios may be constrained due to the restricted data diversity. To enhance the applicability of our method in a wider range of contexts, it would be necessary to incorporate more diverse datasets. This implies that for future enhancements of this work, gathering and integrating data from various sources or environments should be considered to improve the model's broad applicability. (2) Our method does not interact with the victim model. If we could adjust the generation strategy based on query results, it might enhance the success rate of attacks.

## Ethics Statement

The datasets we use in our paper, Advbench(Chen et al., 2022), TCAB(Asthana et al., 2022) and Founta dataset(Founta et al., 2018), are both open-source. We do not disclose any non-open-source data, and we ensure that our actions comply with ethical standards. However, it is worth noting that our method is highly efficient in generating adversarial examples and does not require access to the victim model. This could potentially be misused by users for malicious activities such as attacking communities, spreading rumors, or disseminating spam emails. Therefore, it is important for the research community to pay more attention to security-related studies.

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

# A  Algorithm of CAWR

We define the rules for word substitutions as $(w \rightarrow \hat{w})$, where $w$ is a word in the original sentence $x$, and $\hat{w}$ represents the corresponding replacement word in the adversarial sentence $\hat{x}$. We propose an algorithm to discover Cross-task Adversarial Word Replacement rules. We employ $h(w \rightarrow \hat{w})$ to denote the significance statistics of the substitution rule, which is measured by the change in model confidence. We employ a large-scale statistical approach to identify highly transferable rules (see Algorithmic 1). After obtaining the rules, we

can achieve text perturbation by sequentially replacing the most significant words until reaching the perturbation threshold.

---

**Algorithm 1** Adversarial rules Extraction

---

**Require:**
  **D**: A set of test datasets $\mathcal{D}_i$ on different tasks $i$
  **M**: A set of models $\mathcal{M}_i$ on different tasks $i$
**Ensure:** a set of word replacement rules as well
  as their salience.

1: **for** each datasets $\mathcal{D}_i$ in **D do**
2:   **for** each instance $(x, y)$ in $\mathcal{D}_i$ **do**
3:     $\hat{x} \leftarrow \text{attack}(x, \mathcal{M}_i)$
4:     $z \leftarrow \text{argmax } \mathcal{M}_i(\hat{x})$
5:     **if** $z \neq y$ **then**
6:       **for** each $\hat{w}$ that replaces $w$ **do**
7:         **if** (synonym($\hat{w}$)∪$\hat{w}$)∩synonym($w$)$\neq \emptyset$ **then**
8:           $c(w_i \rightarrow \hat{w}_i) = c(w_i \rightarrow \hat{w}_i) + 1$
9:           $\Delta = (\mathcal{M}_i(x)[\mathbf{y}] - \mathcal{M}_i(\hat{x})[\mathbf{y}])$
10:          $h(w_i \rightarrow \hat{w}_i) = h(w_i \rightarrow \hat{w}_i) + \Delta$
11: **for** each word replacement rule **do**
12:   $h(w \rightarrow \hat{w}) = h(w \rightarrow \hat{w})/c(w \rightarrow \hat{w})$

---

## B  Defense

The following are the defense effects on the CG-Fake and Hatebase datasets. Here, Query refers to the average number of attack queries for successful attack samples, not the average of the total number of queries.

| Method | CGFake | | Hatebase | |
|---|---|---|---|---|
| | ASR(%) | Query | ASR(%) | Query |
| TextFooler | 18.2 | 360.13 | 10.4 | 78.46 |
| +defend | 16.25 | 396.25 | 10.0 | 78.63 |
| PWWS | 69.0 | 489.78 | 9.9 | 107.20 |
| +defend | 54.0 | 481.72 | 9.0 | 105.13 |
| BERT-Attack | 94.6 | 400.61 | 56.8 | 139.14 |
| +defend | 92.75 | 462.92 | 56.7 | 152.27 |
| SememePSO(maxiter=20) | 52.25 | 236.34 | 66.0 | 88.07 |
| +defend | 48.75 | 240.34 | 65.25 | 92.71 |
| DeepWordBug(power=5) | 41.7 | 75.00 | 56.4 | 21.40 |
| +defend | 1.25 | 80.23 | 1.25 | 21.07 |
| DeepWordBug(power=25) | 68.8 | 75.28 | 85.4 | 21.72 |
| +defend | 1.5 | 80.23 | 2.5 | 21.08 |
| DeepWordBug(power=100) | 93.0 | 81.14 | 90.0 | 21.96 |
| +defend | 2.25 | 80.24 | 2.25 | 21.08 |

Table 7: Attack success rate of preprocessing defense using a generator trained with adversarial samples and without defense strategies.