# OpenReview forum: "CT-GAT: Cross-Task Generative Adversarial Attack based on Transferability"
_EMNLP/2023/Conference — EMNLP 2023 Main_

### Official Review · Reviewer_sMHo · 2023-08-05

**Soundness:** 3

**Excitement:**

4: Strong: This paper deepens the understanding of some phenomenon or lowers the barriers to an existing research direction.

**Paper Topic And Main Contributions:**

This paper aims to reduce the dependence of adversarial attack methods on the information of victim models. The authors propose to directly construct adversarial samples by transferable features extracted from various tasks. Specifically, using adversarial samples constructed in different tasks, the author trained a seq-to-seq model called CT-GAT to generate universal adversarial samples and conducted experiments on new tasks. The experiments indicate that the proposed methods have good attack performance.

**Questions For The Authors:**

Question A:
Are different tasks trained in parallel?

Question B:
What are the losses designed for different tasks？

Question C:
How to balance the losses between different tasks?

Question D:
How does the temperature of the model works?

**Reasons To Accept:**

1. The core idea of the method is intuitive and easy to follow.

2. The experimental results are promising.

**Reasons To Reject:**

1. There are a few formatting errors.
Table 3 is marked as Fig 3 erroneously.

2. The details of the proposed method are missing.
In this paper, the authors only mentioned the use of the seq-to-seq model (e.g., DAST) to build a generation model, but did not explain details of the generation model in the training process, such as the loss function used in different tasks. It is better to have a more detailed explanation of Fig 2.

3. The authors mentioned “temperature” several times, and it can affect the performance of the model. However, the authors did not provide a concrete example to explain it.

**Reproducibility:**

3: Could reproduce the results with some difficulty. The settings of parameters are underspecified or subjectively determined; the training/evaluation data are not widely available.

**Reviewer Confidence:**

3: Pretty sure, but there's a chance I missed something. Although I have a good feel for this area in general, I did not carefully check the paper's details, e.g., the math, experimental design, or novelty.

---

> ### Author Rebuttal · Authors · 2023-08-29
>
> We sincerely appreciate your keen eye in identifying the typo error in our paper. We will diligently review our work to identify and rectify any similar issues. In the revised version, we commit to thoroughly correcting all spelling errors. Your attention to detail is invaluable in enhancing the quality of our work. Thank you once again for your feedback.
>
> **Question A**: Are different tasks trained in parallel?
>
> **Question B**: What are the losses designed for different tasks?
>
> **Question C**: How to balance the losses between different tasks?
>
> **Answer A, B, C:**
>
> Our method does not involve parallel training of different tasks. Instead, we combine adversarial examples derived from various tasks to form a unified training dataset. We refer to our approach as "cross-task", as we generate adversarial examples based on tasks that are distinct from the target task. This is a novel approach compared to previous transferability-based methods, which only conducted transfers within the same task.
>
> For training our adversarial sample generator, we utilize cross entropy, a common sequence-to-sequence loss function.
>
> Since our method does not incorporate multi-task training, there is no need for a strategy to balance the losses between different tasks.
>
> We hope this clarifies your concerns, and we are open to further discussions if needed.
>
> **Question D**: How does the temperature of the model works?
>
> **Answer D**:
>
> The "temperature" in our generative model is a hyperparameter that controls the randomness of predictions during the generation process. In the context of our method, a higher temperature value implies a higher degree of perturbation in the generated adversarial samples, which in turn leads to a higher attack success rate.
>
> We have conducted a series of experiments to illustrate the impact of temperature on the attack success rate. The results, as shown in Table A, depict the variation in attack success rate with different temperature values. We hope this clarifies your question, and we are open to further discussions if needed.
>
> **Table A.** The impact of temperature on the attack success rate(ASR%) of CT-GAT attacks on different victim models.
>
> | Dataset\| Temperature | 1.0  | 1.1  | 1.3  | 1.5  | 1.7  | 1.9  | 2.1  | 2.3  | 2.5  |
> | --------------------- | ---- | ---- | ---- | ---- | ---- | ---- | ---- | ---- | ---- |
> | CGFake                | 49.3 | 58.3 | 72.8 | 81.3 | 92.3 | 98.3 | 100  | 100  | 100  |
> | jigsaw                | 36.8 | 45.3 | 50.0 | 56.8 | 60.0 | 63.5 | 65.3 | 67.5 | 69.0 |
>
> We assure that we will include a line graph illustrating the impact of temperature on the attack success rate in the appendix of the revised version.
>
>
>
> **Reply to your reasons to reject**
>
> I greatly appreciate your pointing out our typos. In the revised version, we will carefully review and correct spelling errors and citation mistakes in the article.
>
> And thank you for pointing out the issue with Figure 2 and its related description not being clear enough.
>
> Figure 2 is the flowchart of our model's training and usage process. In the revised version, we will redraw Figure 2, remove one of the loss boxes, and make it clearer. At the same time, we will add more details in Section 3.2, "Transferability-Based Generator," to explain Figure 2.
>
> **Additional remarks**
>
> We greatly appreciate your time and effort in reviewing our work. We would like to highlight that our proposed method is designed to be straightforward and easy to implement. For the convenience of others to follow our work, once accepted, we will make our code, datasets, and trained model parameters all publicly available on both GitHub and Google Cloud.

---

### Official Review · Reviewer_sPVz · 2023-08-08

**Soundness:** 2

**Excitement:**

2: Mediocre: This paper makes marginal contributions (vs non-contemporaneous work), so I would rather not see it in the conference.

**Paper Topic And Main Contributions:**

This paper proposes a model titled 'CT-GTA' to generate Cross-Task Generative Adversarial Attacks across various NLP tasks. CT-GTA is claimed to exhibit high transferability and can produce universal adversarial attacks for diverse tasks. Experiments have been conducted to demonstrate the performance of CT-GTA.

**Reasons To Accept:**

1. The research topic of "generating universal attacks across different NLP tasks" is interesting and important.
2.  Many datasets and metrics have been deployed for evaluation purposes.

**Reasons To Reject:**

1. The objective of Figure 1 remains unclear, and its intended demonstration needs further clarification.

2. The technical aspects of CT-GAT are inadequately explained, making it challenging to grasp the workings and the rationale behind its effectiveness.

3. Figure 2 is difficult to comprehend and could benefit from improvements to enhance its clarity.

4. The paragraph in Line 199-208 lacks essential details, such as clearly specifying the two tasks being referred to.

5. The explanation regarding how CT-GAT ensures transferability is unclear and requires more elaboration.

6. Section 4.6 "Case Study" is insufficiently developed, leaving readers confused about its relevance and purpose in the overall context.

**Reproducibility:**

1: Could not reproduce the results here no matter how hard they tried.

**Reviewer Confidence:**

4: Quite sure. I tried to check the important points carefully. It's unlikely, though conceivable, that I missed something that should affect my ratings.

---

> ### Author Rebuttal · Authors · 2023-08-29
>
> **Question 1.** The objective of Figure 1 remains unclear, and its intended demonstration needs further clarification.
>
> **Answer 1.**
>
> We appreciate your feedback regarding the clarity of Figure 1. We understand that its objective was not adequately explained, and we're grateful for the opportunity to provide further clarification.
>
> Figure 1, the heatmap presented in Section 5.2 on replacement preferences, illustrates the frequency with which replaced words are substituted with different synonyms.
>
> Previous methods based on transferability used a greedy approach to select the synonym with the highest frequency for substitution[1]. However, our heatmap reveals that, in addition to the highest frequency synonyms, there are other synonyms with high frequencies. The greedy approach could limit the diversity of adversarial samples, thereby constraining the effectiveness of the method.
>
> By utilizing generative models to learn from both the original and adversarial samples, we can learn these transferable features and acquire a more diverse set of replacement rules. This is one of the factors that enhance the transferability of our method.
>
> We commit to providing more detailed descriptions about Figure 1 in the revised version of our paper. We hope this clarifies your concerns and we are open to further discussions if needed.
>
>
>
> [1] On the Transferability of Adversarial Attacks against Neural Text Classifier (Yuan et al., EMNLP 2021)
>
>
>
> **Question 2.** The technical aspects of CT-GAT are inadequately explained, making it challenging to grasp the workings and the rationale behind its effectiveness.
>
> **Answer 2.**
>
> Our methodology offers a novel approach to the community for executing decision-based black-box attacks. Our method bolsters transferability by leveraging adversarial features, specifically word substitution rules, across various tasks.
>
> We have delved into the source factors of transferability and the potential of learning with generative models from three perspectives:
>
> 1. **Cross-task Word Adversarial Replacement (CWAR) rules:** We further studied the methods utilizing transferability and proposed Cross-task Word Adversarial Replacement (CWAR) rules, effectively improving transferability.
> 2. **Similar vulnerable words across different models**: We identified that vulnerable words in sentences are prime targets for attacks. Many white-box attacks exploit this, leading to these vulnerable parts of sentences being more likely to be replaced. Our research has shown that different model structures exhibit similar patterns of vulnerable word distribution, a factor not utilized by previous transferability methods. Our analytical experiments have validated that our approach, which involves learning sequence generation from original to adversarial text, is capable of capturing the distribution patterns of vulnerable words that are closely aligned with those in the victim model.
> 3. **Replacement preference**: Through the Gini imbalance coefficient and intuitive reflection of the heat map, we discovered a greater diversity in replacement patterns. This is evident in the statistical rules of original and adversarial samples. We aim to capture this using a Seq-to-Seq structure. In contrast, previous methods that leveraged transferability only capitalized on replacement patterns with the highest frequency of occurrence.
>
> Our method integrates these three factors, effectively employing certain transferability rules, enabling it to be task and model agnostic during attacks. This proves particularly effective for some black-box scenarios, especially attacks on tasks lacking substitute models, thereby offering a fresh approach to black-box attacks.
>
> For more specific details and quantitative analysis, we kindly invite you to refer to the 'Further Analysis' section (Section 5) of our paper. We hope this clarifies your concerns and we are open to further discussions if needed.
>
> **Question 3.** Figure 2 is difficult to comprehend and could benefit from improvements to enhance its clarity.
>
> **Answer 3.**
>
> We appreciate your feedback regarding the clarity of Figure 2.
>
> Figure 2 is a flowchart that outlines the training and application process of our method.
>
> Specifically, the training process involves taking 'original sample-adversarial sample' pairs from different tasks, mixing these samples together, and learning using a seq-to-seq model. After training, the model can directly generate multiple adversarial samples for an original sample without needing feedback from the victim model. These adversarial samples will be used directly to attack the victim model.
>
> In the revised version of our paper, we will modify Figure 2 to more clearly illustrate our process.
>
>
>
> **Question 4.** The paragraph in Line 199-208 lacks essential details, such as clearly specifying the two tasks being referred to.
>
> **Answer 4.**
>
> Thank you for pointing out the unclear parts of our description.
>
> The section of the paper you're referring to mentions that our training set is derived from two classification tasks: abuse detection and sentiment analysis. Abuse detection is a task that involves identifying aggressive content in text, such as attacks, insults, cyberbullying, hate speech, and so on [1]. Sentiment analysis, on the other hand, is a task used to detect emotions or sentiments, such as happiness, sadness, anger, and so forth [2].
>
> In the revised version of our paper, we will make it clear that both are classification tasks and provide a more detailed introduction to the objectives of each task.
>
>
>
> [1] Abusive language detection in online user content(Nobata et al., WWW 2016)
>
> [2] Sentiment Analysis Based on Deep Learning: A Comparative Study(Dang et al., ICCSEA 2022)
>
>
>
> **Question 5.** The explanation regarding how CT-GAT ensures transferability is unclear and requires more elaboration.
>
> **Answer 5.**
>
> Our approach builds upon previous research on explainability, while also introducing our unique innovations.
>
> Past research found that the frequency of word replacement somewhat reflects the degree of transferability. This was determined by summarizing the phenomenon of transferability, which led to the proposal of a greedy replacement strategy[1].
>
> Our method introduces new perspectives, specifically, that transferability can be enhanced by better leveraging the following three aspects: the frequency of replacement across different tasks, the similarity of vulnerable words across different models, and the diversity in the choice of replacement words.
>
> 1. **Similar vulnerable words across different models:** Our experiments revealed that different models share similar patterns of vulnerable word distribution. These vulnerable words are reflected in the choice of replaced words across various attack methods. The diversity of replacement words can be determined through statistical analysis.
> 2. **Replacement preference:** Our experimental results indicate that the selection frequency of replacement words is unevenly distributed, and there are multiple optimal replacement words. This aspect was not considered in previous methods based on transferability.
> 3. **Transferability across different tasks:** We expanded previous methods to include cross-task transferable attacks and proposed the Cross-Task Word Adversarial Replacement (CWAR) method. Our method uses word replacement frequency from various tasks, not just from the same task or dataset as the victim model. Tests have shown that using word replacement frequency from different tasks enhances transferability.
>
> Our generator model's task is to learn how to perform word replacement searches in the original sample by integrating the three transferability-influencing factors mentioned above. Our method effectively uses transferability rules by merging these three factors. Compared to the baseline, our attack method demonstrates superior performance, indicating that our method has effectively learned transferability.
>
> For more specific details and quantitative analysis, we kindly invite you to refer to the 'Further Analysis' section (Section 5) of our paper. We hope this provides the clarity you were seeking and we are open to further discussions if needed.
>
>
>
> [1] On the Transferability of Adversarial Attacks against Neural Text Classifier (Yuan et al., EMNLP 2021)
>
>
>
> **Question 6.** Section 4.6 "Case Study" is insufficiently developed, leaving readers confused about its relevance and purpose in the overall context.
>
> **Answer 6.**
>
> We appreciate your feedback regarding the development of Section 4.6 "Case Study".
>
> The primary objective of our case study is to analyze and illustrate the adversarial examples generated by our method. The perturbation patterns of these adversarial examples are derived from previous attack methods, effectively learning the characteristics of these prior adversarial examples. Furthermore, our model can also integrate different attack methods, such as character-level attacks and word-level attacks.
>
> In the revised version of our paper, we will provide a more detailed explanation about our case study.
>
>
>
> **Additional remarks**
>
> We greatly appreciate your time and effort in reviewing our work. We would like to highlight that our proposed method is designed to be straightforward and easy to implement. For the convenience of others to follow our work, once accepted, we will make our code, datasets, and trained model parameters all publicly available on both GitHub and Google Cloud.

---

### Official Review · Reviewer_UDzM · 2023-08-10

**Typos Grammar Style And Presentation Improvements:** There is a typo in the abstract (L17)
**Soundness:** 4

**Excitement:**

4: Strong: This paper deepens the understanding of some phenomenon or lowers the barriers to an existing research direction.

**Paper Topic And Main Contributions:**

The authors describe a new method of performing black box adversarial attacks by training a generative model to convert a query into an adversarial query. They confirm the efficacy of the method by training their generative model on a set of adversarial queries generated using standard methods and performing attacks on BERT models finetuned for 10 different tasks.

The proposed method is very query efficient after training while also having a high attack success rate.

**Reasons To Accept:**

A. The method is interesting as it proposes that adversarial queries are task and model agnostic for the most part. This can be leveraged to create generative models that are very efficient at attacking other models.

B. The evaluations are thorough:
- Show examples of the generated samples
- Compare across 10 datasets with different tasks

C. The method works black box - training weights are not required

**Reasons To Reject:**

A small, albeit minor point, is that the model used for finetuning is kept fixed. It could be possible that this attack only works because of some inherent adversarial weakness in BERT's weights which leads to commonality in failure modes across finetuned versions

**Reproducibility:**

5: Could easily reproduce the results.

**Reviewer Confidence:**

2: Willing to defend my evaluation, but it is fairly likely that I missed some details, didn't understand some central points, or can't be sure about the novelty of the work.

---

> ### Author Rebuttal · Authors · 2023-08-29
>
> We are sincerely grateful for your recognition of our work and for your insightful comments. Your feedback is invaluable to us. We also appreciate your diligence in pointing out the writing errors. Rest assured, we will meticulously review the entire manuscript, identifying and rectifying any errors in the text. Thank you once again for your thoughtful and constructive feedback.
>
> **Question 1.**  The model used for finetuning is kept fixed. It could be possible that this attack only works because of some inherent adversarial weakness in BERT's weights which leads to commonality in failure modes across finetuned versions.
>
> **Answer 1.**
>
> We appreciate your point about the potential inherent adversarial weakness in BERT's weights leading to commonality in failure modes across fine-tuned versions. To address this, we have conducted additional tests using larger models to verify the applicability of our method beyond BERT-based classifiers. As shown in Table A, our method still achieves a high attack success rate and requires fewer query times, even with the generally higher robustness of larger models. This aligns with current research findings on the robustness of larger models. The specific results and implementation details are as follows.
>
> **Table A.** The attack performance of the CT-GAT method on GPT-3.5-turbo.
>
> | Task             | Misinformation                                               | Disinformation                                         | &nbsp;&nbsp;&nbsp;&nbsp;&nbsp;&nbsp;&nbsp;&nbsp;Toxic | &nbsp;&nbsp;&nbsp;&nbsp;&nbsp;&nbsp;&nbsp;Spam        | &nbsp;&nbsp;&nbsp;Sensitive Information             |
> | ---------------- | ------------------------------------------------------------ | ------------------------------------------------------ | ----------------------------------------------------- | ----------------------------------------------------- | --------------------------------------------------- |
> | Method\| Dataset | &nbsp;&nbsp;&nbsp;&nbsp;&nbsp;&nbsp;&nbsp;&nbsp;&nbsp;&nbsp;LUN | &nbsp;&nbsp;&nbsp;Amazon-LB                            | &nbsp;&nbsp;&nbsp;&nbsp;&nbsp;&nbsp;&nbsp;Founta      | SpamAssassin                                          | &nbsp;&nbsp;&nbsp;&nbsp;EDENCE                      |
> |                  | ASR(%)\|query                                                | ASR(%)\|query                                          | ASR(%)\|query                                         | ASR(%)\|query                                         | ASR(%)\|query                                       |
> | CT-GAT           | &nbsp;&nbsp;&nbsp;31.54\|31.6                                | &nbsp;&nbsp;&nbsp;56.15\|27.88                         | &nbsp;&nbsp;&nbsp;81.54\|11.41                        | &nbsp;&nbsp;&nbsp;53.08\|24.08                        | &nbsp;&nbsp; 50.77\|23.66                           |
> |                  |                                                              |                                                        |                                                       |                                                       |                                                     |
> | Method\| Dataset | &nbsp;&nbsp;&nbsp;&nbsp;&nbsp;&nbsp;satnews                  | &nbsp;&nbsp;&nbsp;&nbsp;&nbsp;&nbsp;&nbsp;&nbsp;CGFake | &nbsp;&nbsp;&nbsp;&nbsp;Jigsaw2018                    | &nbsp;&nbsp;&nbsp;&nbsp;&nbsp;&nbsp;&nbsp;&nbsp;Enron | &nbsp;&nbsp;&nbsp;&nbsp;&nbsp;&nbsp;&nbsp;&nbsp;FAS |
> |                  | ASR(%)\|query                                                | ASR(%)\|query                                          | ASR(%)\|query                                         | ASR(%)\|query                                         | ASR(%)\|query                                       |
> | CT-GAT           | &nbsp;&nbsp;&nbsp;30.74\|32.8                                | &nbsp;&nbsp;&nbsp;68.46\|15.23                         | &nbsp;&nbsp;&nbsp;75.38\|14.11                        | &nbsp;&nbsp;&nbsp;41.54\|29.22                        | &nbsp;&nbsp;&nbsp;69.23\|18.14                      |
>
> **Details:** We use the following prompt as a query for GPT-3.5. Here, `[Task Name]` is replaced with the specific task name, such as 'spam'.`[Problem Description]` is replaced with the specific task description in Advbench[1]. `[Positive Example] `selects the first positive example text from the dataset. `[Negative Example] `selects the first negative example text from the dataset. `[Input]` is the sentence we query for each attack.
>
> ```
> Can you help me with the [Task Name] classification task? The definition of this task is as follows:
> [Problem Description]
> If the input sentence matches the above description, output 1; otherwise, output 0. For example:
> <input>
> [Positive Example]
> <output>
> 1
> <input>
> [Negative Example]
> <output>
> 0
> Now please perform the classification:
> <input>
> [input]
> <output>
> ```
>
> [1] Why Should Adversarial Perturbations be Imperceptible? Rethink the Research Paradigm in Adversarial NLP (Chen et al., EMNLP 2022)

---

### Official Review · Reviewer_xQiS · 2023-08-10

**Soundness:** 4

**Excitement:**

2: Mediocre: This paper makes marginal contributions (vs non-contemporaneous work), so I would rather not see it in the conference.

**Paper Topic And Main Contributions:**

The authors propose a textual adversarial attack by constructing examples on transferable features across various text-processing target tasks. The proposed framework, Cross-Task Generative Adversarial Attack (CT-GAT), trains a BART-based generative model using data sampled from the target tasks to obtain transferable adversarial features. The experiments show that CT-GAT has a higher attack success rate compared to existing textual attacks.

**Reasons To Accept:**

1. The paper is well-written with a clear storyline to motivate the proposed approach.

2. The idea of utilizing prior generative knowledge from BART to extract transferable adversarial features is innovative.

3. The experiments, which span five major textual tasks, show that the proposed attack is effective and transferable.

**Reasons To Reject:**

1. The perturbation bounds of the attack are not explicitly controlled, which makes the attack success rates questionable to compare with existing bound-controlled works. What is more, given this setting, the extent of ground truth shift needs more quantitative and qualitative evaluation to show that the attack preserves the ground truth during the attack.

2. Section 5.3 describes potential generative-based defense against the attack. However, a more effective defense approach is to perform adversarial training on the target model to robustify the prediction. Authors should consider discussing more effective defense approaches rather than preprocessing/filtering.

3. The five target tasks (Misinformation, Disinformation, Toxic, Spam, Sensitive Information) described by the paper can be conducted by Large Langauge Model (LLM) + In-context Learning (e.g., show examples to LLM and let LLM perform classification). The transferability and effectiveness of the adversarial feature for these LLMs remain questionable. Authors should consider discussing such real-world applicability for the work.

4. Table 4 evaluates the transferability of the attack. The paper should add additional quantitative analysis between the transferability and attack success rates.

**Reproducibility:**

3: Could reproduce the results with some difficulty. The settings of parameters are underspecified or subjectively determined; the training/evaluation data are not widely available.

**Reviewer Confidence:**

3: Pretty sure, but there's a chance I missed something. Although I have a good feel for this area in general, I did not carefully check the paper's details, e.g., the math, experimental design, or novelty.

---

> ### Author Rebuttal · Authors · 2023-08-29
>
> **Question 1.** The perturbation bounds of the attack are not explicitly controlled, which makes the attack success rates questionable to compare with existing bound-controlled works. What is more, given this setting, the extent of ground truth shift needs more quantitative and qualitative evaluation to show that the attack preserves the ground truth during the attack.
>
> **Answer 1.**
>
> Thank you for your insightful suggestion. In existing generative adversarial methods, controlling the perturbation boundary is a challenge. To address this, we trained our adversarial sample generator using traditional adversarial methods with controlled boundaries, such as limiting the perturbation rate. This approach aids the model in learning boundary rules.
>
> In our paper, we have evaluated the perturbation rate of our method using metrics like edit distance, grammatical errors, and increase rate in perplexity (as shown in Table 5). The results indicate that our method, compared to baseline methods with explicit boundary control, exhibits a relatively lower perturbation rate.
>
> Regarding the preservation of semantic truth during the attack, we have conducted experiments (detailed in Table A). Our method shows promising semantic preservation capabilities, ranking in the upper-middle level among existing methods that limit perturbation boundaries. The specific experimental data and settings are as follows:
>
> **Table A.** Semantic Similarity Measurement Table. ('ASR' represents Attack Success Rate (%), 'cos' represents cosine similarity of sentence vectors, 'GPT' represents the score evaluated by GPT-3.5-turbo)
>
> | Task                | &nbsp;&nbsp;Disinformation                                   | &nbsp;&nbsp;&nbsp;&nbsp;&nbsp;&nbsp;&nbsp;&nbsp;&nbsp;&nbsp;&nbsp;&nbsp;&nbsp;&nbsp;Toxic | Sensitive Information                                        |
> | :------------------ | ------------------------------------------------------------ | ------------------------------------------------------------ | ------------------------------------------------------------ |
> | Method \| Dataset   | &nbsp;&nbsp;&nbsp;&nbsp;&nbsp;&nbsp;&nbsp;Amazon-LB          | &nbsp;&nbsp;&nbsp;&nbsp;&nbsp;&nbsp;&nbsp;&nbsp;&nbsp;&nbsp;&nbsp;&nbsp;Founta | &nbsp;&nbsp;&nbsp;&nbsp;&nbsp;&nbsp;&nbsp;&nbsp;&nbsp;&nbsp;&nbsp;EDENCE |
> |                     | ASR(%)\| cos \|GPT                                           | ASR(%)\| cos \|GPT                                           | ASR(%)\| cos \|GPT                                           |
> | TextFooler          | &nbsp;&nbsp;&nbsp;&nbsp;&nbsp;&nbsp;9.0   \|**0.95**\|**4.57** | &nbsp;&nbsp;&nbsp;&nbsp;52.7 \|**0.87**\|3.01                | &nbsp;&nbsp;&nbsp;&nbsp;23.9 \|**0.91**\|3.18                |
> | PWWS                | &nbsp;&nbsp;&nbsp;&nbsp;18.8 \|0.84\|3.74                    | &nbsp;&nbsp;&nbsp;&nbsp;61.0 \|0.81\|2.49                    | &nbsp;&nbsp;&nbsp;&nbsp;46.0 \|0.87\|3.09                    |
> | BERT-Attack         | &nbsp;&nbsp;&nbsp;&nbsp;43.0 \|0.81\|2.24                    | &nbsp;&nbsp;&nbsp;&nbsp;77.0 \|0.76\|3.35                    | &nbsp;&nbsp;&nbsp;&nbsp;**90.3** \|0.74\|2.93                |
> | PSO(maxiter=100)    | &nbsp;&nbsp;&nbsp;&nbsp;23.8 \|0.92\|3.92                    | &nbsp;&nbsp;&nbsp;&nbsp;79.5 \|0.80\|2.64                    | &nbsp;&nbsp;&nbsp;&nbsp;79.6 \|0.86\|2.70                    |
> | Deep(power=5)       | &nbsp;&nbsp;&nbsp;&nbsp;&nbsp;&nbsp;9.3 \|0.82\|4.57         | &nbsp;&nbsp;&nbsp;&nbsp;40.7 \|0.73\|**4.55**                | &nbsp;&nbsp;&nbsp;&nbsp;22.9 \|0.57\|**4.79**                |
> | Deep(power=25)      | &nbsp;&nbsp;&nbsp;&nbsp;12.4 \|0.32\|4.48                    | &nbsp;&nbsp;&nbsp;&nbsp;65.5 \|0.24\|3.40                    | &nbsp;&nbsp;&nbsp;&nbsp;79.9 \|0.08\|4.68                    |
> | ROCKET              | &nbsp;&nbsp;&nbsp;&nbsp;38.7 \|0.71\|1.74                    | &nbsp;&nbsp;&nbsp;&nbsp;97.0 \|0.05\|2.85                    | &nbsp;&nbsp;&nbsp;&nbsp;84.5 \|0.32\|2.88                    |
> | CT-GAT_{word}(ours) | &nbsp;&nbsp;&nbsp;&nbsp;19.5 \|0.90\|3.45                    | &nbsp;&nbsp;&nbsp;&nbsp;90.3 \|0.81\|2.74                    | &nbsp;&nbsp;&nbsp;&nbsp;53.0 \|0.87\|2.89                    |
> | CT-GAT(ours)        | &nbsp;&nbsp;&nbsp;&nbsp;**56.2** \|0.73\|3.41                | &nbsp;&nbsp;&nbsp;&nbsp;**99.3** \|0.80\|2.86                | &nbsp;&nbsp;&nbsp;&nbsp;88.0 \|0.76\|2.93                    |
> |                     |                                                              |                                                              |                                                              |
> | Method \| Dataset   | &nbsp;&nbsp;&nbsp;&nbsp;&nbsp;&nbsp;&nbsp;&nbsp;&nbsp;&nbsp;&nbsp;&nbsp;CGFake | &nbsp;&nbsp;&nbsp;&nbsp;&nbsp;&nbsp;&nbsp;Jigsaw2018         | &nbsp;&nbsp;&nbsp;&nbsp;&nbsp;&nbsp;&nbsp;&nbsp;&nbsp;&nbsp;&nbsp;&nbsp;&nbsp;&nbsp;&nbsp;FAS |
> |                     | ASR(%)\| cos \|GPT                                           | ASR(%)\| cos \|GPT                                           | ASR(%)\| cos \|GPT                                           |
> | TextFooler          | &nbsp;&nbsp;&nbsp;&nbsp;18.2 \|**0.93**\|3.82                | &nbsp;&nbsp;&nbsp;&nbsp;12.5 \|**0.84**\|2.60                | &nbsp;&nbsp;&nbsp;&nbsp;17.4 \|**0.90**\|3.34                |
> | PWWS                | &nbsp;&nbsp;&nbsp;&nbsp;69.0 \|0.80\|3.25                    | &nbsp;&nbsp;&nbsp;&nbsp;20.2 \|0.79\|2.64                    | &nbsp;&nbsp;&nbsp;&nbsp;36.5 \|0.84\|3.37                    |
> | BERT-Attack         | &nbsp;&nbsp;&nbsp;&nbsp;94.6 \|0.79\|2.42                    | &nbsp;&nbsp;&nbsp;&nbsp;40.4 \|0.77\|2.76                    | &nbsp;&nbsp;&nbsp;&nbsp;**92.4** \|0.75\|2.50                |
> | PSO(maxiter=100)    | &nbsp;&nbsp;&nbsp;&nbsp;67.2 \|0.86\|3.13                    | &nbsp;&nbsp;&nbsp;&nbsp;51.9 \|0.76\|2.54                    | &nbsp;&nbsp;&nbsp;&nbsp;61.4 \|0.82\|2.94                    |
> | Deep(power=5)       | &nbsp;&nbsp;&nbsp;&nbsp;41.7 \|0.83\|**4.61**                | &nbsp;&nbsp;&nbsp;&nbsp;35.9 \|0.60\|**3.95**                | &nbsp;&nbsp;&nbsp;&nbsp;40.8 \|0.58\|**4.61**                |
> | Deep(power=25)      | &nbsp;&nbsp;&nbsp;&nbsp;68.8 \|0.26\|3.89                    | &nbsp;&nbsp;&nbsp;&nbsp;57.6 \|0.25\|3.45                    | &nbsp;&nbsp;&nbsp;&nbsp;77.6 \|0.15\|4.50                    |
> | ROCKET              | &nbsp;&nbsp;&nbsp;&nbsp;97.2 \|0.60\|1.82                    | &nbsp;&nbsp;&nbsp;&nbsp;64.2 \|0.05\|1.95                    | &nbsp;&nbsp;&nbsp;&nbsp;82.0 \|0.41\|2.43                    |
> | CT-GAT_{word}(ours) | &nbsp;&nbsp;&nbsp;&nbsp;69.3 \|0.86\|3.36                    | &nbsp;&nbsp;&nbsp;&nbsp;67.0 \|0.76\|2.15                    | &nbsp;&nbsp;&nbsp;&nbsp;42.0 \|0.84\|2.60                    |
> | CT-GAT(ours)        | &nbsp;&nbsp;&nbsp;&nbsp;**99.2** \|0.79\|3.28                | &nbsp;&nbsp;&nbsp;&nbsp;**75.2** \|0.70\|2.48                | &nbsp;&nbsp;&nbsp;&nbsp;86.8 \|0.69\|2.94                    |
>
> **Details:** Referring to previous methods[1], we use the cosine similarity of sentence vectors generated by the "Universal-Sentence-Encoder-4" model to measure the similarity between sentences. Simultaneously, we use the GPT-3.5-turbo model for evaluation, allowing it to give a score from 1 to 5 to measure semantic similarity, with a higher score indicating greater similarity. We use the following prompt as a query for GPT-3.5. `[sentence A]` and `[sentence B]` will be respectively replaced with the original sentence and the adversarial sentence.
>
> ```
> Please provide an integer between 1 and 5 to measure the degree of semantic similarity between Sentence A and Sentence B, with a higher score indicating greater similarity:
> <Sentecne A>
> [sentence A]
> <Sentence B>
> [sentence B]
> <Output>
> ```
>
> Moreover, we have undertaken manual evaluations of sentence semantic similarity, as detailed in the Human Evaluation section of our paper. As evidenced in Table 2, our method successfully preserved the original sentence's meaning in over 88% of the instances. This further underscores the efficacy of our approach in maintaining semantic integrity while generating adversarial examples.
>
>
>
> [1] Grey-box Adversarial Attack And Defence For Sentiment Classification (Xu et al., NAACL 2021)
>
>
>
> **Question 2.** Section 5.3 describes potential generative-based defense against the attack. However, a more effective defense approach is to perform adversarial training on the target model to robustify the prediction. Authors should consider discussing more effective defense approaches rather than preprocessing/filtering.
>
> **Answer 2.**
>
> We appreciate your suggestions. We have utilized traditional adversarial training methods, mixing adversarial samples (including both word-level and character-level) with original samples for model training, based on some adversarial training articles[1,2]. As shown in Table B, while adversarial training proves to be a better defense against word-level attacks, it doesn't always perform well against character-level attacks. Interestingly, our method shows even better performance against character-level attack methods (such as deepwordbug).
>
> **Table B**. Defense effect comparison. "Query" refers to the average number of iterative queries needed for successful adversarial samples. A lower Attack Success Rate (ASR ↓) signifies better defense effects. A higher Query count (Query ↑) also indicates better defense effects.
>
> |    Defend\|Attack    |    TextFooler    |       PWWS       |   BERT-Attack    |         PSO         |   deepwordbug    |
> | :------------------: | :--------------: | :--------------: | :--------------: | :-----------------: | :--------------: |
> |                      | ASR(↓) \| Query(↑) | ASR(↓) \| Query(↑) | ASR(↓) \| Query(↑) |  ASR(↓) \| Query(↑)   | ASR(↓) \| Query(↑) |
> |      No Defend       |    10.4\|78.5    |  9.9\|**107.2**  |   56.8\|139.1    |     66.0\|88.1      |  56.4\|**21.4**  |
> | Adversarial training | &nbsp;&nbsp; **8.6**\|**87.9**   |  **7.4**\|100.7  | **54.9**\|147.3  | **63.1**\|**116.4** |    55.7\|21.2    |
> |    CT-GAT-Defend     |    10.0\|78.6    |    9.0\|105.1    | 56.7\|**152.3**  |     65.3\|92.7      | &nbsp;&nbsp; **1.3**\|21.1   |
>
> [1] Towards a robust deep neural network in texts: A survey (Wang et al., arXiv:1902.07285 2021)
>
> [2] Defense of word-level adversarial attacks via random substitution encoding (Z. Wang and H. Wang, KSEM 2022)
>
>
>
> **Question 3.** The five target tasks described by the paper can be conducted by LLM + In-context Learning. The transferability and effectiveness of the adversarial feature for these LLMs remain questionable. Authors should consider discussing such real-world applicability for the work.
>
> **Answer 3.**
>
> Thank you for your constructive suggestion. To validate the effectiveness of our method, we conducted tests using GPT-3.5-turbo, a large language model (LLM). As shown in Table C, we present the attack success rate and the average number of queries required to obtain an adversarial sample on this LLM. The results suggest that while the large model is generally more robust than the fine-tuned smaller models, its robustness still leaves room for improvement. This observation aligns with current research on the robustness of large models[1]. Despite this, our method still achieves a high attack success rate, indicating that our approach has high transferability and is applicable in real-world scenarios.
>
> **Table C.** The attack performance of the CT-GAT method on GPT-3.5-turbo.
>
> | Task             | Misinformation | Disinformation | &nbsp;&nbsp;&nbsp;&nbsp;&nbsp;&nbsp;&nbsp;&nbsp;Toxic         | &nbsp;&nbsp;&nbsp;&nbsp;&nbsp;&nbsp;&nbsp;Spam          | &nbsp;&nbsp;&nbsp;Sensitive Information |
> | ---------------- | -------------- | -------------- | ------------- | ------------- | --------------------- |
> | Method\| Dataset | &nbsp;&nbsp;&nbsp;&nbsp;&nbsp;&nbsp;&nbsp;&nbsp;&nbsp;&nbsp;LUN            | &nbsp;&nbsp;&nbsp;Amazon-LB      | &nbsp;&nbsp;&nbsp;&nbsp;&nbsp;&nbsp;&nbsp;Founta        | SpamAssassin  | &nbsp;&nbsp;&nbsp;&nbsp;EDENCE                |
> |                  | ASR(%)\|query  | ASR(%)\|query  | ASR(%)\|query | ASR(%)\|query | ASR(%)\|query         |
> | CT-GAT           | &nbsp;&nbsp;&nbsp;31.54\|31.6    | &nbsp;&nbsp;&nbsp;56.15\|27.88   | &nbsp;&nbsp;&nbsp;81.54\|11.41  | &nbsp;&nbsp;&nbsp;53.08\|24.08  |&nbsp;&nbsp; 50.77\|23.66          |
> |                  |                |                |               |               |                       |
> | Method\| Dataset | &nbsp;&nbsp;&nbsp;&nbsp;&nbsp;&nbsp;satnews        | &nbsp;&nbsp;&nbsp;&nbsp;&nbsp;&nbsp;&nbsp;&nbsp;CGFake         | &nbsp;&nbsp;&nbsp;&nbsp;Jigsaw2018    | &nbsp;&nbsp;&nbsp;&nbsp;&nbsp;&nbsp;&nbsp;&nbsp;Enron         | &nbsp;&nbsp;&nbsp;&nbsp;&nbsp;&nbsp;&nbsp;&nbsp;FAS                   |
> |                  | ASR(%)\|query  | ASR(%)\|query  | ASR(%)\|query | ASR(%)\|query | ASR(%)\|query         |
> | CT-GAT           | &nbsp;&nbsp;&nbsp;30.74\|32.8    | &nbsp;&nbsp;&nbsp;68.46\|15.23   | &nbsp;&nbsp;&nbsp;75.38\|14.11  | &nbsp;&nbsp;&nbsp;41.54\|29.22  | &nbsp;&nbsp;&nbsp;69.23\|18.14          |
>
> **Details:** We use the following prompt as a query for GPT-3.5. Here, `[Task Name]` is replaced with the specific task name, such as 'spam'.`[Problem Description]` is replaced with the specific task description in Advbench[2]. `[Positive Example] `selects the first positive example text from the dataset. `[Negative Example] `selects the first negative example text from the dataset. `[Input]` is the sentence we query for each attack.
>
> ```
> Can you help me with the [Task Name] classification task? The definition of this task is as follows:
> [Problem Description]
> If the input sentence matches the above description, output 1; otherwise, output 0. For example:
> <input>
> [Positive Example]
> <output>
> 1
> <input>
> [Negative Example]
> <output>
> 0
> Now please perform the classification:
> <input>
> [input]
> <output>
> ```
>
> [1] How robust is gpt-3.5 to predecessors? a comprehensive study on language understanding tasks. (Chen et al., arXiv:2303.00293  2023)
>
> [2] Why Should Adversarial Perturbations be Imperceptible? Rethink the Research Paradigm in Adversarial NLP (Chen et al., EMNLP 2022)
>
>
>
> **Question 4.** Table 4 evaluates the transferability of the attack. The paper should add additional quantitative analysis between the transferability and attack success rates.
>
> **Answer 4.**
>
> We appreciate your suggestion about adding additional quantitative analysis between the transferability and attack success rates.
>
> For attack methods that are based on transferability, their transferability is typically measured by the attack success rate. For non-transferable attack methods, adversarial samples are first generated on a specific model, and then used to attack other models, with the success rate serving as the measure of effectiveness. Table 4 in our paper presents a comparison of the transferability of our Cross-task Adversarial Word Replacement (CAWR) method on tasks of varying scopes. Since CAWR is a transferable method, the attack success rate is directly used as a measure of its effectiveness. We will make sure to clarify this in the text. The references we used to measure transferability are as follows:
>
> \[1\] On the Transferability of Adversarial Attacks against Neural Text Classifier (Yuan et al., EMNLP 2021)
>
> [2] Learning transferable adversarial perturbations (Krishna kanth Nakka and Mathieu Salzmann, NeurIPS 2021)
>
> [3] Toward understanding and boosting adversarial transfer-ability from a distribution perspective (Zhu et al.,TIP 2022)

---

### Meta-Review · Area_Chair_Qgin · 2023-09-18

**Recommendation:** 4

**Metareview:**

The authors propose a new method for black-box model adversarial attacks. They leverage a generative model to use data sampled from the target tasks to obtain transferable adversarial features. In the experiments, the authors show a high attack success rate across multiple tasks. The paper is well-written and easy to follow.
At the same time, the authors might miss some detailed descriptions of their methods and results.

---

### Decision · Program_Chairs · 2023-10-07

**Decision:**

Accept-Main

**Comment:**

The authors propose a new method for black-box model adversarial attacks. They leverage a generative model to use data sampled from the target tasks to obtain transferable adversarial features. In the experiments, the authors show a high attack success rate across multiple tasks. The paper is well-written and easy to follow.
At the same time, the authors might miss some detailed descriptions of their methods and results.